# Circadian patterns of heart rate, respiratory rate and skin temperature in hospitalized COVID-19 patients

**Harriët M. R. van Goor**[1]*, **Kim van Loon**[2], **Martine J. M. Breteler**[1,2], **Cornelis J. Kalkman**[2], **Karin A. H. Kaasjager**[1]

**1** Department of Acute Internal Medicine, University Medical Center Utrecht, Utrecht, The Netherlands,
**2** Department of Anesthesiology, University Medical Center Utrecht, Utrecht, The Netherlands

* h.m.r.vangoor-3@umcutrecht.nl

## Abstract

### Rationale

Vital signs follow circadian patterns in both healthy volunteers and critically ill patients, which seem to be influenced by disease severity in the latter. In this study we explored the existence of circadian patterns in heart rate, respiratory rate and skin temperature of hospitalized COVID-19 patients, and aimed to explore differences in circadian rhythm amplitude during patient deterioration.

### Methods

We performed a retrospective study of COVID-19 patients admitted to the general ward of a tertiary hospital between April 2020 and March 2021. Patients were continuously monitored using a wireless sensor and fingertip pulse oximeter. Data was divided into three cohorts: patients who recovered, patients who developed respiratory insufficiency and patients who died. For each cohort, a population mean cosinor model was fitted to detect rhythmicity. To assess changes in amplitude, a mixed-effect cosinor model was fitted.

### Results

A total of 429 patients were monitored. Rhythmicity was observed in heartrate for the recovery cohort ($p<0.001$), respiratory insufficiency cohort ($p<0.001$ and mortality cohort ($p = 0.002$). Respiratory rate showed rhythmicity in the recovery cohort ($p<0.001$), but not in the other cohorts ($p = 0.18$ and $p = 0.51$). Skin temperature also showed rhythmicity in the recovery cohort ($p<0.001$), but not in the other cohorts ($p = 0.22$ and $p = 0.12$). For respiratory insufficiency, only the amplitude of heart rate circadian pattern increased slightly the day before (1.2 (99%CI 0.16–2.2, $p = 0.002$)). In the mortality cohort, the amplitude of heart rate decreased (-1.5 (99%CI -2.6- -0.42, $p<0.001$)) and respiratory rate amplitude increased (0.72 (99%CI 0.27–1.3, $p = 0.002$) the days before death.

**Data Availability Statement:** Data are held in the data repository DataverseNL (https://doi.org/10.34894/78HDKJ). Metadata and scripts are freely available. Personal data of patients are available

upon request since a data transfer agreement has to be arranged before this data can be shared. Data request can be made using the 'Contact owner' button in DataverseNL. The request will be send to 3 persons: the corresponding author, the principal investigator, and the data manager of the study. The data manager will guarantee long-term data accessibility.

**Funding:** The author(s) received no specific funding for this work.

**Competing interests:** The authors have declared that no competing interests exist.

## Conclusion

A circadian rhythm is present in heart rate of COVID-19 patients admitted to the general ward. For respiratory rate and skin temperature, rhythmicity was only found in patients who recover, but not in patients developing respiratory insufficiency or death. We found no consistent changes in circadian rhythm amplitude accompanying patient deterioration.

## Introduction

Many elements of human physiology follow a circadian rhythm to anticipate and react to environmental changes throughout the day [1]. Acute disruption of this cycle is associated with immune dysregulation [2], delirium [3] and even mortality at the intensive care unit (ICU) [4, 5]. Hospitalization can contribute to disruption of circadian patterns due to artificial light, noise, (sedative) medication, and the fact that the individual sleep-wake cycle of a patient has to make way for the hospital routine [1]. In addition, the illness itself can cause circadian disruption, for example in the case of systemic inflammation [1, 6–9]. Neuroinflammation and neurodegeneration specifically might alter the regulation genes, or clock genes, responsible for a normal 24-hour cycle. Coronavirus disease 2019 (COVID-19) has several characteristics that may lead to disruption of circadian rhythms. COVID-19 is accompanied by sleep disturbance [10], neuroinflammation [11], and in severe cases systemic inflammation and encephalopathy [12–14]. Since July 2020, patients with COVID-19 are treated with dexamethasone [15], which can affect the circadian pattern of the human metabolism depending on time of administration [16]. Moreover, circadian patterns of heart rate and respiratory rate can be disturbed by acute hypoxia [17], a common symptom of severe COVID-19.

Several vital signs have shown to follow a circadian rhythm [18–20]. Even in critically ill patients admitted to the ICU, where vital signs are highly influenced by medication and ventilation, circadian patterns were found in respiratory rate, heart rate, blood pressure and temperature [21]. Previous research in ICU settings has shown that circadian rhythm becomes increasingly more pronounced in recovering patients (who will eventually be discharged home), as opposed to patients who will not survive or were discharged with palliative care [21]. However, circadian patterns in vital signs thus far have mainly been studied in either healthy volunteers, or in critically ill patients at the ICU (where continuously recorded data is readily available). Since the development of wireless sensors, continuous monitoring of vital signs at the general hospital ward has become more common [22]. Data can be used for visual monitoring by clinicians, and for the development of clinical decision support models, to detect deterioration of patients at an earlier stage. However, the alarm strategies of many systems are mainly based on single threshold breaches. Aspects of vital sign trends, like a circadian pattern, are not considered, even though incorporating vital signs trends has the potential to improve prediction models and alarm strategies considerably [23, 24]. Moreover, changes in circadian patterns themselves could be valuable predictors of deterioration. A recent study used changes in circadian rhythm characteristics to identify SARS-CoV-2 infection and predict COVID-19 diagnosis [25].

In this exploratory study, we aimed to answer three related research questions. First, we assessed whether circadian rhythms can be observed for heart rate, respiratory rate and skin temperature in COVID-19 patients admitted to a general hospital ward. Subsequently, we assessed to what extent these circadian rhythms exist in patients who develop respiratory insufficiency, patients who died, and patients who recovered without developing respiratory

insufficiency. Lastly, we explored whether changes in the amplitude of circadian rhythms of vital signs can be observed in deteriorating patients, and could therefore be possible predictors of deterioration.

## Methods

We performed a retrospective cohort study of patients who were diagnosed with COVID-19. Patients were offered the chance to opt-out of retrospective data analyses during hospital registration and again at hospital discharge, according to the institutional protocol. A waiver for ethical review was obtained from the medical ethical research committee Utrecht (MERC-20-365). The study was conducted according to the principles of the Declaration of Helsinki and the General Data Protection Regulation [26, 27].

### Setting

During the pandemic, a continuous wireless monitoring system for vital signs was deployed at the COVID-19 cohort ward of a tertiary medical center in Utrecht, the Netherlands, starting April 1, 2020. This system recorded heart rate, respiratory rate and skin temperature twice per minute, using a wearable wireless patch sensor (Biosensor Voyage, Philips Electronics Netherlands BV) and peripheral oxygen saturation (SpO2) via a finger pulse-oximeter (EarlyVue VS30, Philips Electronics Netherlands BV). The patch sensor was attached on the left hemithorax, approximately 2 cm sub clavicular, and was replaced every three days following manufacturer instructions. Patients with a pacemaker did not receive a sensor since ECG-derived respiratory rate measurements are unreliable in paced rhythms. Heart rate, respiratory rate and oxygen saturation was real-time available for all caregivers to support care. The values for skin temperature were not directly available, since the clinical relevance of skin temperature is unsure and not yet integrated in general hospital care.

### Data collection

Patients were included starting April 1, 2020 until March 1, 2021. Inclusion was stopped because the manufacturer stopped delivering these sensors to focus on the production of other sensors, but the replacement did not meet the accuracy requirements. All patients with confirmed COVID-19 and available continuous sensor data were included. To be able to describe the cohort, baseline characteristics were recorded from the electronic patient record, including the Charlson Comorbidity Index for predicting 1-year mortality [28].

### Data selection

Patients were divided into three groups: patients who recovered without experiencing respiratory insufficiency, and patients with severe clinical deterioration, divided into patients who developed respiratory insufficiency and patients who died. We chose these three groups since respiratory insufficiency and mortality are both outcomes of severe patient deterioration, but follow a different course. Patients seldom died unexpectedly, and often received palliative care in the last days before death. Therefore we decided to analyze this group separately. If a patient developed respiratory insufficiency at any point during admission, he or she was included in the respiratory insufficiency cohort, and not in the recovered cohort. If a patient developed respiratory insufficiency and died while being monitored, he or she was included in the mortality cohort instead of the respiratory insufficiency. Respiratory insufficiency was defined as the need for 15 l/min oxygen therapy, high flow oxygen therapy or mechanical ventilation, whichever came first. We did not deem ICU admission a suitable endpoint since a substantial

part of the population had treatment restrictions preventing them from ICU admission, and the hospital regularly struggled with capacity problems at the ICU. Instead, we chose the endpoint hypoxic respiratory insufficiency, which better reflects the starting point of severe illness in COVID-19. The time and date of onset of respiratory insufficiency was manually collected from the electronic patient record.

Since the length of stay and length of continuous monitoring varied among patients, we chose to only include 3 days (72 hours) of data for each patient. This way we aimed to avoid overrepresentation of patients with more data. For patients in the respiratory insufficiency cohort, we selected the 72 hours before onset of respiratory insufficiency. For patients who died, we selected the 72 hours of data preceding death. Since respiratory insufficiency usually occurred within the first 72 hours (median 33 hours) of admission, we selected the first 72 hours of data for patients in the recovery group as a comparable control. Since at least 4 hours of data was needed for statistical analysis, patients with less than 4 hours of continuous data in the selected 72-hour timeframe were excluded.

All continuous vital sign data was validated before use: physiologically improbable data was removed using a predefined computer algorithm. Since our cohort included dying patients, we used wide limits for improbable data (for respiratory rate <1/min & >80/min; for heart rate <30/min & >280/min; for skin temperature < 25˚C). Artifacts in respiratory rate and heart rate were filtered by removing large abrupt changes that lasted for less than 2 minutes (for respiratory rate a change of >20/min, for heart rate a change of >25/min). To ensure we only used skin temperature data of periods that the wearable was attached to the patient, and not the data of the preparation period, we only used skin temperature data between the first and last valid heart rate measurements. The first 10 measurements (5 minutes) of skin temperature data of each patient were removed, since the sensor needed several minutes to warm up.

## Statistical analysis

To limit the impact of short-lasting outliers and minutes with missing data further, the median of each vital sign per fifteen-minute segment was calculated for each patient. Subsequently we calculated the overall mean of these medians, including a 95% confidence interval (CI) and the 95% upper and lower limit of all measurements. Data was plotted for visual evaluation. For quantitative evaluation we made use of a cosinor model. A cosinor model is a type of non-linear model used to asses repetitive patterns, such as circadian rhythms [29]. A cosinor consists of several components. The MESOR (midline estimating statistic of rhythm) is the rhythm adjusted mean of the modelled variable, e.g. the rhythm adjusted mean heart rate. The amplitude is the measure of the extent of predictable change within the cycle, e.g. 2 heart beats/min. Two times the amplitude is the difference between the highest and lowest point of the cosinor regression line. The acrophase represents the timing of overall high values in a cycle, expressed in (negative) degrees, where the reference time is set to 0˚, and a full period is 360˚. The period is the (expected) duration of one cycle, which is 24 hours for circadian cycles. For this study, we fitted two separate cosinor models. First, we used a cosinor model of the population mean to estimate the mean coefficients of the three cohorts and to detect rhythmicity, using R package 'cosinor2' [29]. This model illustrates mean differences between the cohorts. Rhythmicity was determined by the fit of the cosinor model using the F-ratio. However, this model does not account for correlation within individual patients and cannot assess longitudinal changes in data. Therefore, we fitted a cosinor mixed effects model as second model, using R package 'cosinoRmixedeffects' [25, 30]. This allows for random MESOR, amplitude and acrophase per patient. We included an interaction term with the day on which measurements were taken, to see if coefficients changed over the three-day observation period. To estimate means and mean

differences, we used a bootstrapping method with 500 simulations [30]. For more elaborate explanation of this method we refer to the article by Hirten et al. [25]. A p-value of 0.01 was deemed to be statistically significant for quantitative analysis. R software version 4.0.3 (R foundation for Statistical Computing, Vienna, Austria 2021) was used for all analyses.

## Results

Between April 1$^{st}$ 2020 and March 1$^{st}$ 2021, a total of 429 COVID-19 patients were continuously monitored at the ward. Of these, 368 could be included for analysis: 296 patients who recovered without developing respiratory insufficiency, 27 patients who died, and 45 patients who developed respiratory insufficiency and either recovered, or died without being monitored (Fig 1). Table 1 shows a description of the cohort. Note that patients who died were older, had more comorbidities, received dexamethasone less often and had a higher rate of 'Do not ventilate' orders.

### Assessment of rhythmicity

Fig 2 shows the raw overall mean of the vital signs in the three cohorts. Both the respiratory insufficiency and mortality cohort had a small sample size and wide confidence intervals. Rhythmicity in mean heart rate was found in all cohorts (recovery p<0.001, respiratory insufficiency p<0.001, mortality p0.002) (Table 2). Rhythmicity in mean respiratory rate and mean skin temperature was only found in the recovery cohort (resp. p<0.001 and p<0.001).

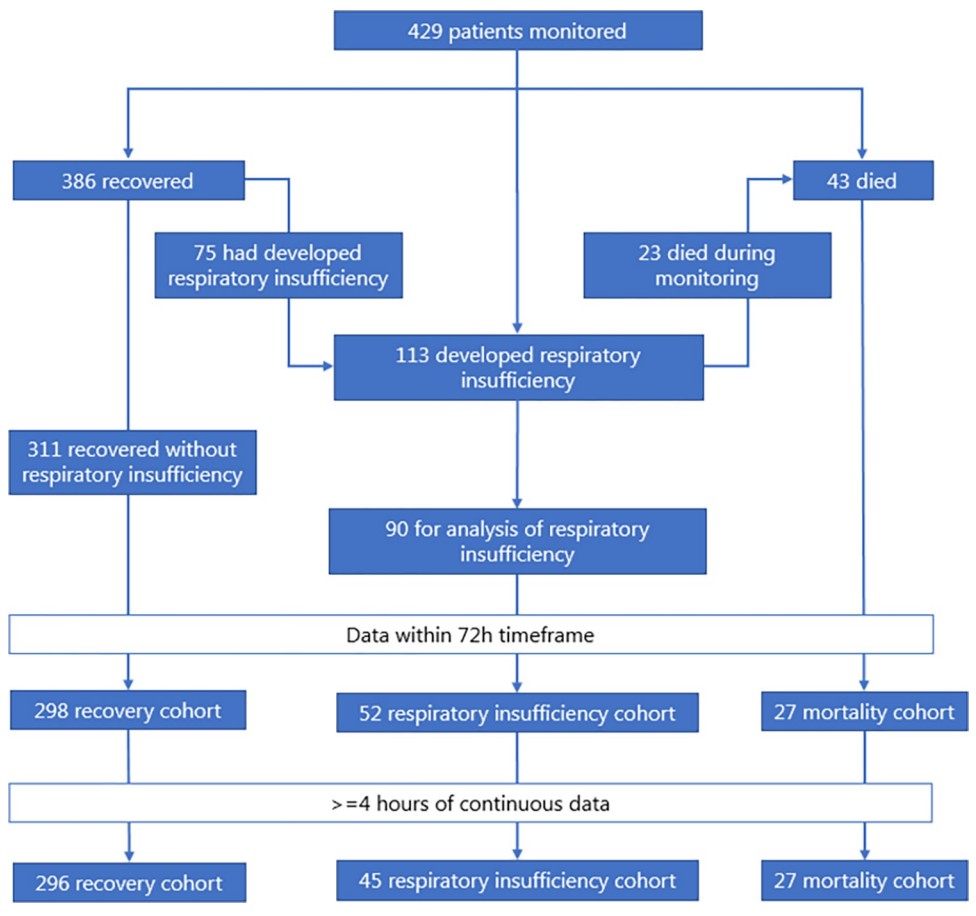

**Fig 1. Flowchart of patient inclusion and data selection.**

**Table 1. Patient characteristics and median duration of recorded vital signs during three-day observation period.**

| | | *All* | *Recovery* | *Resp. insuf.* | *Mortality* |
|---|---|---|---|---|---|
| Number of patients | | 368 | 296 | 45 | 27 |
| Age (median, IQR) | | 65 (55–74) | 63.5 (55–72) | 64 (56–73) | 76 (71–82) |
| Male sex (n, %) | | 221 (60.0%) | 181 (61.1%) | 25 (55.6%) | 15 (55.6%) |
| CCI (median, IQR) | | 3 (1–4) | 2 (1–4) | 3 (2–4) | 4 (4–6) |
| Dexamethasone administration (n, %) | | 279 (75.8%) | 223 (75.3%) | 38 (84.4%) | 18 (66.7%) |
| 'Do not ventilate' order (n, %) | | 91 (24.7%) | 57 (19.3%) | 10 (22.2%) | 24 (88.9%) |
| Length of stay (median days, IQR) | | 7 (4–11) | 6 (4–10) | 15 (10–31) | 8 (5–13) |
| Median (IQR) hours of data per patient during 72-hour timeframe | • Heart rate | 72 (46.8–72) | 72 (60–72) | 34 (25.8–70.3) | 68.8 (26.4–72) |
| | • Respiratory rate | 62.1 (38.3–72) | 63.5 (48.9–72) | 31.5 (18.7–51.9) | 60.5 (17.3–72) |
| | • Skin temperature | 63.6 (37.8-63-6) | 72 (52.5–72) | 30.8 (12.4–51.9) | 60 (15–72) |

Resp. insuf.: hypoxic respiratory insufficiency, CCI: Charlson Comorbidity Index based on 1 year mortality, IQR: interquartile range

## Changes in circadian pattern amplitude

The cosinor characteristics for each cohort per day are presented in Fig 3 and S1 Fig. The MESOR values for heart rate and respiratory rate were lower in the recovery cohort than the

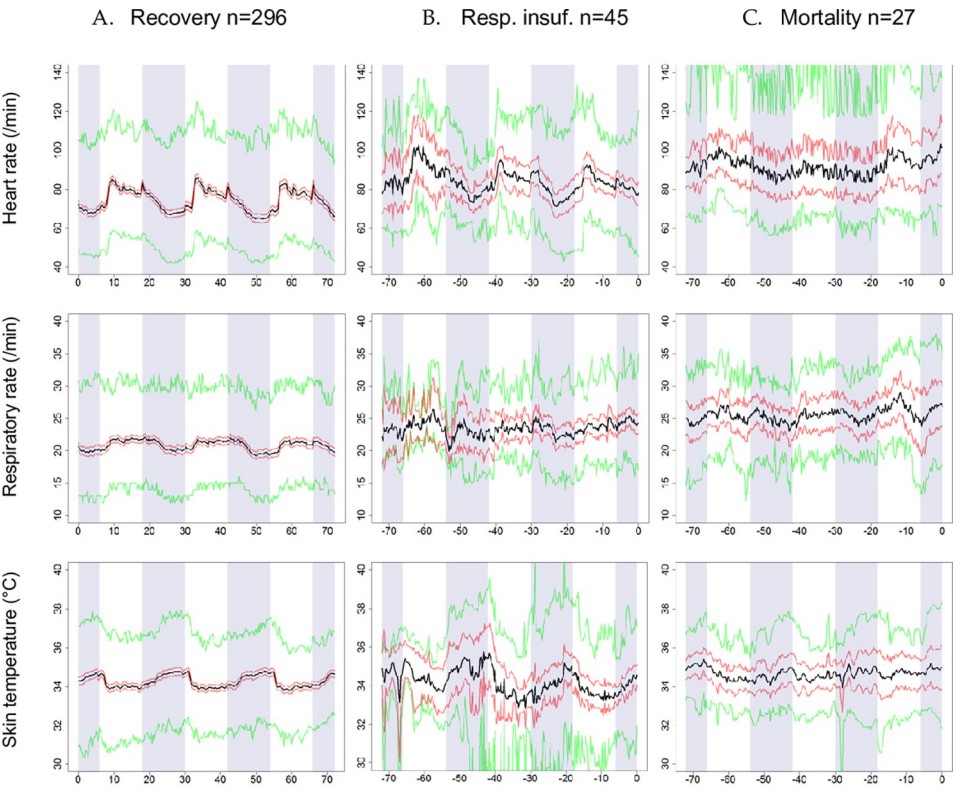

X-axis: hours after first day of admission (A), before day of respiratory insufficiency (B) or before day of death (C). White panels represent time between 06.00-18.00 and blue panels represent time between 18.00-06.00. Resp. insuf.: respiratory insufficiency. Black: overal mean. Red: 95%CI of overal mean. Green: upper and lower limit of 95% range of all means.

**Fig 2. Mean of vital signs during three day observation period in each cohort.**

**Table 2. Coefficients of cosinor models.** Recovered patients are compared to patients with respiratory insufficiency and deceased patients.

|  | Recovered (95%CI) | Resp. insuf. (95%CI) | p-value of difference | Died (95%CI) | p-value of difference |
|---|---|---|---|---|---|
| Heart rate (/min) |  |  |  |  |  |
| • MESOR | 74.7 (73.3–76.1) | 78.9 (73.9–84.0) | 0.04 | 95.3 (88.0–102.5) | <0.001 |
| • Amplitude | 6.9 (6.4–7.5) | 5.1 (3.1–7.1) | 0.76 | 4.0 (2.0–5.9) | 0.58 |
| ➢ Rhythmicity | p<0.001 | p<0.001 |  | p = 0.002 |  |
| Respiratory rate (/min) |  |  |  |  |  |
| • MESOR | 20.7 (20.3–211) | 22.7 (21.0–24.5) | 0.001 | 26.0 (24.4–27.6) | <0.001 |
| • Amplitude | 1.0 (0.7–1.2) | 1.4 (-0.24–2.9) | 0.90 | 1.0 (-0.86–3.0) | <0.001 |
| ➢ Rhythmicity | p<0.001 | p = 0.18 |  | p = 0.51 |  |
| Skin temperature (˚C) |  |  |  |  |  |
| • MESOR | 34.2 (34.1–34.3) | 33.2 (31.4–34.8) | 0.003 | 34.6 (34.1–35.1) | 0.07 |
| • Amplitude | 0.39 (0.28–0.50) | 1.5 (-0.2–3.2) | 0.66 | 0.32 (0.02–0.62) | 0.95 |
| ➢ Rhythmicity | p<0.001 | p = 0.22 |  | p = 0.12 |  |

Resp. insuf.: respiratory insufficiency, MESOR: midline estimation statistic of oscillation

respiratory insufficiency cohort, but higher in the mortality cohort. In the recovery cohort, an increase of amplitude was seen for all parameters over the course of the three days. The amplitude for heart rate significantly increased on day 2 (difference of 0.90 (99%CI 0.64–1.2, p<0.001)) and from day 2 to 3 (difference of 0.53 (99%CI 0.21–0.85, p<0.001)) (Table 3). Respiratory rate amplitude increased from day 2 to 3 (difference of 0.25 (99%CI 0.14–0.35, p<0.001)) and skin temperature amplitude increased from day 1 to 2 (difference of 0.10 (99% CI 0.06–0.13, p<0.001). For the respiratory insufficiency cohort, only heart rate showed a clear increase in amplitude (difference day 2 to day 3 of 1.2 (0.16–2.2, p = 0.002)). Skin temperature amplitude initially decreased (difference day 1 to 2 of -0.31 (99%CI -0.48- -0.14, p<0.001)) and later increased (difference day 2 to 3 of 0.16 (99%CI 0.00–0.23, p = 0.006). In the mortality cohort, heart rate amplitude decreased from day 1 to 2 (difference of -1.5 (99%CI -2.6- -0.42, <0.001), and respiratory rate amplitude increased from day 2 to 3 (difference of 0.72 (99%CI 0.27–1.3, p = 0.002).

## Discussion

In patients admitted with COVID-19, we could confirm the presence of a circadian rhythm of heart rate. For respiratory rate and skin temperature, a circadian pattern could only be observed in patients who ultimately recovered. The amplitude of heart rate circadian rhythm increased slightly the day before respiratory insufficiency. In dying patients, a slight decrease in heart rate amplitude and an increase in respiratory rate amplitude can be observed in the days before death. Although statistically significant, these differences were small.

The existence of a circadian rhythm in vital signs has been well established [18–20]. However, in daily clinical practice, this physiological rhythm is hardly considered. With the advent of wireless continuous vital signs monitoring, patterns in vital signs are gaining attention. A recent study on cardiovascular changes in COVID-19 found a repetitive pattern in cardiovascular parameters and hypothesized this to be part of a circadian rhythm [31]. Our study confirms the existence of a circadian pattern in vital signs of hospitalized COVID-19 patients. A study performed in multiple intensive care units demonstrated circadian patterns for blood pressure, heart rate, respiratory rate and temperature [21]. This study found that the difference between the peak and nadir of vital signs is reduced in patients who died compared to patients who recovered. This led to the hypothesis that a decrease in circadian rhythm amplitude might

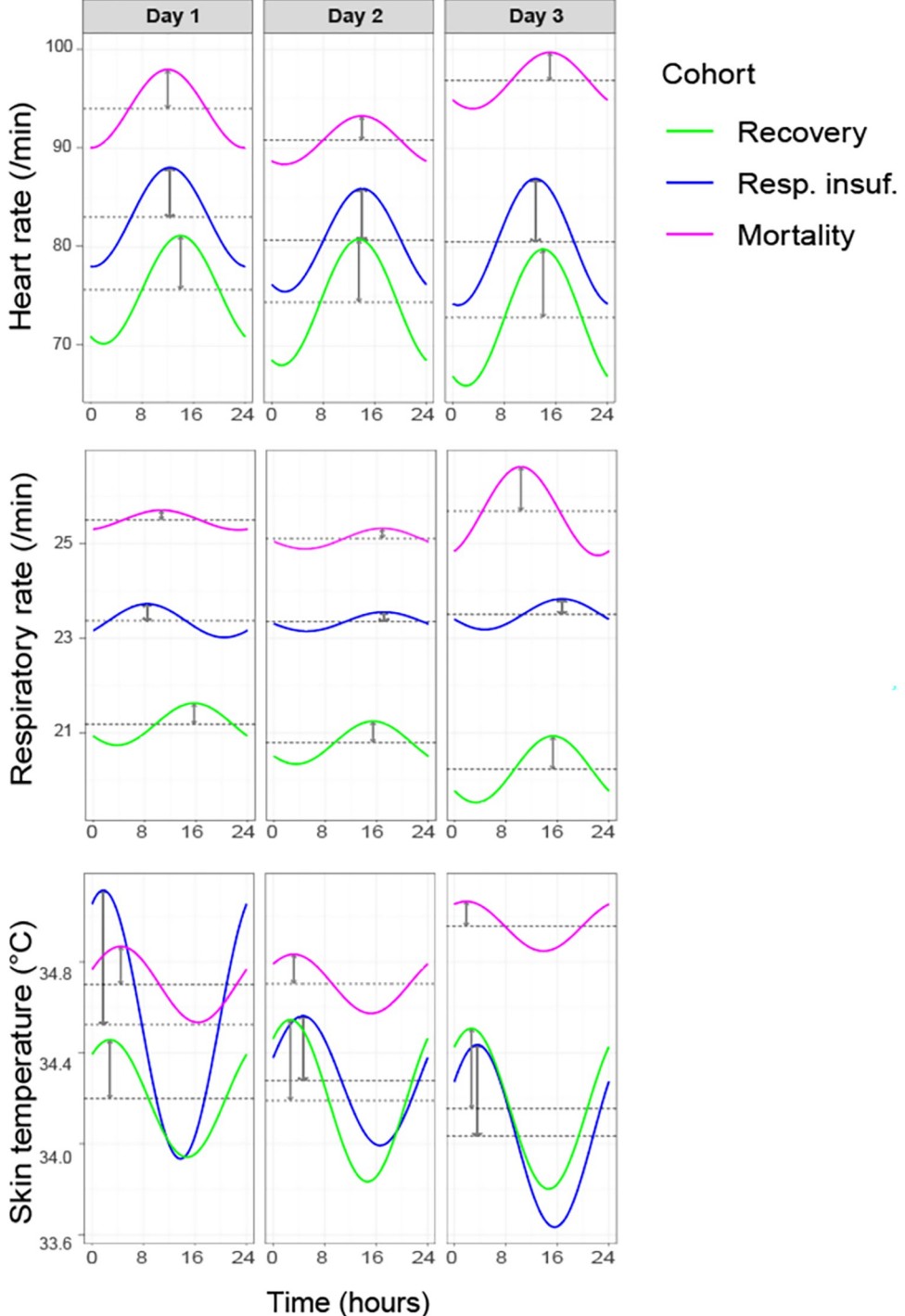

**Fig 3. Progression of cosinor characteristics over the course of three days for heart rate, respiratory rate and skin temperature, stratified by cohort.**

contain prognostic information. In our study, we could not confirm a consistent decrease of circadian rhythm amplitude in deteriorating COVID-19 patients. Some vital signs even showed a slight increase of circadian pattern amplitude during of the observation period. The

**Table 3. Differences in cosinor mixed effect model amplitudes (difference, 99%CI) between days for the A. recovery cohort, B. respiratory insufficiency cohort, and C. mortality cohort.**

| | *day 1 vs day 2* | *p-value* | *day 2 vs day 3* | *p-value* |
|---|---|---|---|---|
| A. Recovery | | | | |
| • Heart rate | 0.90 (0.64–1.2) | <0.001 | 0.53 (0.21–0.85) | <0.001 |
| • Respiratory rate | 0.01 (-0.08–0.10) | 0.82 | 0.25 (0.14–0.35) | <0.001 |
| • Skin temperature | 0.10 (0.06–0.13) | <0.001 | 0.00 (-0.04–0.03) | 0.80 |
| B. Respiratory insufficiency | | | | |
| • Heart rate | 0.20 (-1.2–1.5) | 0.71 | 1.2 (0.16–2.2) | 0.002 |
| • Respiratory rate | -0.15 (-0.60–0.26) | 0.39 | 0.12 (-0.19–0.49) | 0.36 |
| • Skin temperature | -0.31 (-0.48- -0.14) | <0.001 | 0.16 (0.00–0.23) | 0.006 |
| C. Mortality | | | | |
| • Heart rate | -1.5 (-2.6- -0.42) | <0.001 | 0.40 (-0.75–1.6) | 0.39 |
| • Respiratory rate | 0.01 (-0.34–0.35) | 0.96 | 0.72 (0.27–1.3) | 0.002 |
| • Skin temperature | -0.04 (-0.19–0.09) | 0.51 | -0.02 (-0.16–0.11) | 0.68 |

method we used here, however, is different. In the study by Davidson et al., the peak-nadir excursion was used to quantify the circadian rhythm, which is somewhat different from the cosinor amplitude and might be more influenced by temporary peaks and troughs. These methodological differences might explain the observed differences in results.

Although we did not find a decrease in amplitude values for deteriorating patients, we did find a lack of rhythmicity in mean respiratory rate and mean skin temperature in the days leading up to respiratory insufficiency or death. This could be a sign of a generalized disturbed circadian rhythm in these patients. Changes in heart rate and respiratory rate during the day are mostly caused by changes in arousal and level of muscle activity, independent of the time of day [19, 32–34]. If patients are active during the night, e.g. due to severe illness and/or delirium, they could have similar vital signs during these periods as during the day, resulting in a lack of rhythmicity. Periods of fever and hypoxemia could also result in temporary deviations in heart rate and respiratory rate, disrupting the circadian pattern even further.

Patients who died at the hospital ward showed no rhythmicity of respiratory rate and skin temperature, and a decrease of heart rate amplitude two days before death. This is in accordance with the observations of Davidson et al. 2021 [21]. The decrease of circadian rhythm might be caused by several factors. Severe illness has shown to influence clock gene expression and melatonin excretion [35, 36]. Older age is also accompanied with lower levels of melatonin [37]. Comorbidities and medication suppressing the regulation of vital signs, such as metoprolol, could have influenced circadian patterns too. Furthermore, circadian rhythms are influenced by light input [37]. As part of palliative care, patients were often relocated to single rooms with closed blinds for comfort. These patients also often received sedative medication such as opioids and benzodiazepines, blurring the difference between wake and sleep. This might have played a role in the lack of rhythmicity in this cohort. Lastly, patients often died after more than 72 hours of admission. The selected data therefore represents a later part of the admission than the data of the other two cohorts. The longer hospitalization time might have added to the disruption of circadian rhythm. In dying patients, continuous monitoring was often discontinued as part of palliative care too, so unfortunately only few patients could be included for analysis.

Skin temperature showed a circadian pattern opposite from heart rate and respiratory rate, with its peak at night instead of during the day. Core temperature usually drops during the night due to an increase of skin temperature and the subsequent excess heat loss [38–40]. This

is, however, only true for distal body parts. In our study, we used a sensor that was attached to the chest, two centimeter sub clavicular. In such a proximal location, the skin temperature is expected to follow the same pattern of the core temperature [38], instead of the inversed pattern that we observed. Why this phenomenon occurred is unknown.

## Strengths and limitations

This study shows that a circadian rhythm of vital signs is present in hospitalized COVID-19 patients. All patients were admitted with the same disease, with a known pattern of deterioration, and for each patient a large set of data points was available for analysis. This made it possible to not only look at the differences between cohorts, but also to analyze more closely the changes of amplitude during deterioration. Our study also has multiple limitations. Even though the overall sample size was large, the respiratory insufficiency and mortality cohorts were relatively small, resulting in wide confidence intervals. The data selection of the mortality cohort was of a later stage of admission than the other two cohorts, introducing 'hospitalization time' as a possible confounder. In future research, this could be avoided by using a case-control design matched by length of hospital admission. Although previous studies have shown differences in vital signs patterns between men and women [21, 31, 41], we decided not to do a sub analysis based on sex due to the limited sample size of two of the cohorts. Secondly, all patients in our study were admitted with COVID-19, and therefore conclusions can only be drawn regarding this specific population. Lastly, skin temperature can be modified by many factors, including environmental temperature, clothing, showering, and exercise. The effect of miscellaneous factors, such as leakage of airflow from underneath an oxygen mask, are unknown. The clinical relevance and interpretation of skin temperature therefore is uncertain. Nonetheless, the observation that a circadian rhythm is present for skin temperature in COVID-19 patients who recover could be a valuable continuously measured vital parameter for the future.

## Use in predictive modelling and clinical practice

Continuous monitoring is used increasingly outside high care units in an effort to detect deterioration timely [22]. In COVID-19 too, the trajectory of vital signs is hypothesized to aid in the detection of respiratory and cardiovascular decline [31]. Predictive models and alarm strategies could help clinicians to recognize deterioration, without producing too many false alarms [42]. The performance of these models might be influenced by the existence of a circadian rhythm. Previous research has already shown that accounting for differences in vital signs values between day and night may reduce alarm rate in various models at the general ward [24]. The next step in predictive modeling with continuous data is trend analysis, since changes of vital signs might be better predictors than single values [43, 44]. Both model builders and hospital professionals should be aware however that a rise in heart rate and respiratory rate in the morning, or a rise of skin temperature in the evening, might not be a deteriorating trend at all, but rather a part of a physiological rhythm. Even though this should be accounted for, changes in circadian rhythm themselves are unlikely to be useful as predictors of deterioration Lack of rhythmicity is not reflected in a decrease of amplitude, so a different metric should be used to express decrease of rhythmicity. Furthermore, one would need at least 24 hours' worth of data before being able to assess a circadian pattern. Future research should focus on adequately predicting deterioration with vital sign trends despite the existence of circadian patterns. In clinical practice, several general wards have already implemented continuous monitoring for COVID-19 patients [31, 45, 46]. Alarm strategies and escalation protocols are often based on early warning scores, which could be influenced by physiological changes in vital signs over

the day. Based on our clinical experience during the pandemic, the early warning scores of the majority of COVID-19 patients increase in the morning when patients become physically active. Awareness of the existence of a circadian rhythm in common vital signs might aid nurses and physicians in the interpretation of continuous data and continuous early warning scores.

In conclusion, a circadian rhythm is present in heart rate of COVID-19 patients admitted to the general ward. For respiratory rate and skin temperature, rhythmicity was only found in patients who recovered, but not in patients developing respiratory insufficiency or death. We found no consistent changes in circadian rhythm amplitude accompanying patient deterioration.

## Supporting information

**S1 Checklist. STROBE statement—checklist of items that should be included in reports of *cohort studies*.**
(DOCX)

**S1 Fig. Progression of cosinor characteristics per vital sign per group, mean with 99%CI.** MESOR: midline estimation statistic of rhythm. MESOR and amplitude are in /min for heart rate and respiratory rate, and ˚C for skin temperature. Acrophase in (degree).
(TIF)

## Author Contributions

**Conceptualization:** Harriët M. R. van Goor, Kim van Loon, Cornelis J. Kalkman.

**Data curation:** Harriët M. R. van Goor, Martine J. M. Breteler.

**Formal analysis:** Harriët M. R. van Goor.

**Investigation:** Harriët M. R. van Goor, Martine J. M. Breteler.

**Methodology:** Harriët M. R. van Goor, Kim van Loon.

**Project administration:** Harriët M. R. van Goor.

**Resources:** Karin A. H. Kaasjager.

**Supervision:** Kim van Loon, Cornelis J. Kalkman, Karin A. H. Kaasjager.

**Validation:** Harriët M. R. van Goor.

**Visualization:** Harriët M. R. van Goor.

**Writing – original draft:** Harriët M. R. van Goor.

**Writing – review & editing:** Harriët M. R. van Goor, Kim van Loon, Martine J. M. Breteler, Cornelis J. Kalkman, Karin A. H. Kaasjager.

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
