## [Decision Letter · Decision Letter 0]

20 Dec 2021

PONE-D-21-34906Circadian patterns of heart rate, respiratory rate and skin temperature in hospitalized covid-19 patientsPLOS ONE

Dear Dr. van Goor,

Thank you for submitting your manuscript to PLOS ONE. After careful consideration, we feel that it has merit but does not fully meet PLOS ONE’s publication criteria as it currently stands. Therefore, we invite you to submit a revised version of the manuscript that addresses the points raised during the review process.

Fix issues with citations.

Clarify information on dexamethasone treatment and absent data.

Provide more details on methodology and statistics as specified by the reviewers.

Improve discussion about alternative methodologies and include more recent references.

Consider including a study population that did not develop hypoxemic respiratory insufficiency – or at least discuss this omission and tone down conclusions.

Consider including a flow diagram to describe the total number of potentially eligible patients, and those excluded at each stage (e.g. as suggested by the STROBE statement). – 

Avoid including patient data in more than one time-period “cohort” and then comparing differences between time 

If no cosinor analysis is applied, thenit seems like this peak-nadir measurement (PNex) will only provide the range of daytime vs. nighttime data. Please include references for using this measurement in circadian analyses. 

Include a quantitative analysis for rhythmicity.

Discuss more specifically how findings could be translated into the clinics.

We look forward to receiving your revised manuscript.

Kind regards,

Henrik Oster, Ph.D.

Academic Editor

PLOS ONE

Journal Requirements:

Additional Editor Comments:

n/a

Reviewers' comments:

Reviewer's Responses to Questions

**Comments to the Author**

1. Is the manuscript technically sound, and do the data support the conclusions?

Reviewer #1: Yes

Reviewer #2: Partly

Reviewer #3: Partly

2. Has the statistical analysis been performed appropriately and rigorously? 

Reviewer #1: I Don't Know

Reviewer #2: No

Reviewer #3: No

3. Have the authors made all data underlying the findings in their manuscript fully available?

Reviewer #1: Yes

Reviewer #2: Yes

Reviewer #3: Yes

4. Is the manuscript presented in an intelligible fashion and written in standard English?

Reviewer #1: Yes

Reviewer #2: Yes

Reviewer #3: Yes

5. Review Comments to the Author

Reviewer #1: Harriët MR van Goor and colleagues reported the existence of circadian patterns in heart rate, respiratory rate and skin temperature of hospitalized Covid-19 patients. In their study, they compared various stages of disease. Albeit the predictive power of circadian pattern amplitude for disease severity was low, the authors state that accounting for circadian patterns might improve general monitoring- and alarm strategies.

The overall writing style and the accuracy of language is sufficient. However, the novelty of the reported findings is questionable as most of the findings are not specific for Covid-19 and already reproduced several times in other cohorts. Moreover, additional improvements within methods and discussion addressing the recent literature are required.

1. Introduction: “Since September 2020, patients with covid-19 are treated with dexamethasone[15], which has an suppressive effect on the circadian pattern of the human metabolism[16].” The citation (16) is not applicable. In the cited study dexamethasone was administered in the afternoon. Within the clinical routine, dexamethasone is likely to be administered in the early morning which might rather result in a strengthening of circadian rhythms. How many patients received dexamethasone in this study? Why September 2020, not June 2020?

2. Methods: “Inclusion of patients stopped because the wearable sensor was no longer available.” Please specify: Was the availability of the sensor tied to funding for the study or did the company stop production due to unreliable measurement accuracy?

3. Methods: “Patients with a pacemaker did not receive a sensor since RR measurements might result unreliable in paced rhythms.” Please clarify: Were those patients completely excluded from the study or only RR measurements were excluded for those patients?

4. Methods: „Since our cohort included dying patients, we used wide limits for improbably 114 data (for RR <1/min & >80/min; for HR <30/min & >280/min; for sT < 25°C).” The lower limits for temperature and respiratory rate are extremely wide and should be critically revised.

5. Methods: “Data was divided in five cohorts based on different stages of disease…. Hypoxic respiratory insufficiency was defined as the need for 15 l/min oxygen therapy.” Why did the authors not stratify according to the WHO criteria in mild, intermediate and severe COVID19. Please discuss and reference, if this method has been used before.

6. Methods: “For quantitative assessment we divided the data in daytime (06:00-00:00) and nighttime (00:00-06:00).” The daytime period is proportionally much longer than the nighttime. Please discuss and reference, why this method was chosen and if this method has been published before.

7. Methods: Data collection included the Charlson Comorbidity Index, but the group differences were not further discussed within the manuscript. The use of further disease severity scores for ICU patients (SOFA, GCS) would complement the author’s analysis.

8. Statistics: Although the authors nicely removed several abrupt deviations before analysis, the use of PNex measurement might not be picking up circadian trends very well. It is correlated, however results are very noisy. For a better understanding the analysis needs to be discussed and compared with other methods applied in previous publications examining vital signs. Why did the authors not perform a regular rhythmicity analysis and sine curve fit to better estimate the amplitude?

9. Discussion: Studying a clinical cohort of Covid-19 patients exclusively, the authors conclude that a general knowledge of circadian patterns might improve general monitoring- and alarm strategies. This affirmation is not supported by data including other disease entities and therefore needs to be discussed including more recent literature and trials examining circadian patterns in clinical cohorts.

Daou M, Telias I, Younes M, Brochard L, Wilcox ME. Abnormal Sleep, Circadian Rhythm Disruption, and Delirium in the ICU: Are They Related? Front Neurol. 2020 Sep 18;11:549908. doi: 10.3389/fneur.2020.549908. PMID: 33071941

Lachmann G, Ananthasubramaniam B, Wünsch VA, Scherfig LM, von Haefen C, Knaak C, Edel A, Ehlen L, Koller B, Goldmann A, Herzel H, Kramer A, Spies C. Circadian rhythms in septic shock patients. Annals of Intensive Care. 2021 11: 64. PMID 33900485

Maas MB, Lizza BD, Abbott SM, Liotta EM, Gendy M, Eed J, Naidech AM, Reid KJ, Zee PC. Factors Disrupting Melatonin Secretion Rhythms During Critical Illness. Crit Care Med. 2020 Jun;48(6):854-861. PMID: 32317599

Maas MB, Iwanaszko M, Lizza BD, Reid KJ, Braun RI, Zee PC. Circadian Gene Expression Rhythms During Critical Illness. Crit Care Med. 2020 Dec;48(12):e1294-e1299. PMID: 33031153

Reviewer #2: PLOS ONE – D 21 34906

Circadian patterns of HR, RR, and skin temp in hospitalized COVID19 patients

This study evaluated circadian patterns in patients with COVID19 admitted to inpatient ward, using pulseox + “wireless sensor” measurements over 5 different time windows of their hospitalized illness. As a descriptive study this is interesting and provides rationale for continuing to investigate the utility of such measurements for risk prediction and patient monitoring.

The main issue is that the current study design does not support an analysis of risk for hypoxemic respiratory insufficiency as the authors propose. Evaluating different time periods a single cohort of patients, comparing a pre-hypoxemic “control” period and post-hypoxemic “outcome” period in those who developed hypoxemic respiratory insufficiency (defined in this study as >15L O2 therapy), we are able to establish only that these measurements differ in these two periods of time within these patients. To evaluate for risk of hypoxemic respiratory insufficiency, the authors must include a study population that did not develop hypoxemic respiratory insufficiency. This could be done in a variety of ways to create either a cohort or a case-control study design. The analysis should be revised to include such a cohort, or the manuscript needs to be rewritten with this significant change in mind.

It would also be helpful to include references using the methods employed (daytime peak vs. nighttime nadir of physiologic data) to evaluate circadian patterns in the data.

Abstract

- Please define “delta day” as a measurement in the abstract.

- Please revise to more accurately describe the logistic regression results (that VS changes are associated with worsened hypoxemia within in this select cohort of patients; they do not demonstrate risk of developing hypoxemia – as above, this cannot be evaluated unless the study also includes patients who did not develop hypoxemic respiratory insufficiency.)

Methods

- p5. Please clarify - if sT measurements were not directly available then how were they studied?

- Please include how many COVID19 patients were excluded for lack of sensor data.

- p.6. Please correct “improbably data.”

- The authors excluded patients with < 48 hours of continuous data during a three-day period, except for the post-hypoxemic respiratory failure period. This suggests that patients who developed hypoxemic respiratory failure very quickly (eg after 1-2 days) or who died shortly after admission without developing hypoxemic respiratory failure, etc. would have been excluded. This biases the study population to those patients who had a relatively slow progression of disease and may bias the study population away from those admitted to the ICU (it is unclear whether remote monitoring continued after ICU admission?). The authors should consider revising the analysis, using a shorter time window for each “period”.

- Consider including a flow diagram to describe the total number of potentially eligible patients, and those excluded at each stage (e.g. as suggested by the STROBE statement).

- Including patient data in more than one time-period “cohort” and then comparing differences between time periods is problematic from a study design perspective. This introduces bias towards the null – which makes it more difficult to identify true differences in vital signs between cohorts. The authors should consider revising their compared time periods so that no overlap occurs.

- How accurately was the onset of hypoxemic respiratory insufficiency defined? It would be helpful to describe how this information was obtained, e.g. if this was time stamped (hour and minute) to ensure that the vital sign data was assigned correctly as pre- vs. post-intervention.

- p.7. Were peaks and nadirs drawn from raw data? Typically, circadian analyses using peak/nadir are drawn from a cosinor or sinusoidal analysis that accounts for the entire dataset shifting + and – in a circadian fashion, which is less sensitive to random noise (e.g., Shoben Am J Epi 2011 for an excellent example.) If no cosinor analysis is applied, thenit seems like this peak-nadir measurement (PNex) will only provide the range of daytime vs. nighttime data. Please include references for using this measurement in circadian analyses.

Results:

- p. 7. Please rename the 5 time periods; they are not truly “cohorts” as they include the same patients, and the patient data is actually duplicated across some of these analytic groups. Suggest using “time periods” or something similar.

- How do more patients have respiratory insufficiency than deterioration? This might be clarified by using more distinct/separated definitions of time periods for comparison and/or a flow diagram as suggested above.

- Table 3 sugests that day 1 vs. 2 of respiratory insufficiency and day 1 vs 3 of respiratory insufficiency were compared in logisitc regression. This is different from the analyses described in the Methods. Please clarify.

-p.9; was mean PNex compared quantitatively between stages using a statistical test? If so, consider adding these comparisons to table 2.

-p.9, lines 189-191 and 191-192 should be edited for accuracy. Since patients without hypoxemic respiratory insufficiency were not included in this analysis, we cannot evaluate whether this development was associated with the development of hypoxemia. All we know is that the delta PNex was associated with skin temperature in this select cohort of patients who will go on to develop respiratory insufficiency.

-Table 3: Table 3 compares the difference in Pnex between days 1 and 2 and days 1 and 3. Please clarify – is this limited to stage III patients? If so, please add this detail and the number of patients included to the title.

Discussion:

- I agree that Figure 1 suggests qualitatively that there is circadian variability given the cyclical nature of overall VS pattern. However, it’s hard to claim definitively that there is no circadian variability based on a subjective assessment (eg for the patients with mortality.) From figure 1, stage I and stage III do not necessarily look all that different – it is not clear what the authors used as a cut off for circadian rhythm presence vs. absence. If a quantitative analysis is possible, one should be included. If not, the discussion should reflect this uncertainty throughout.

- The Discussion should be edited in several places to accurately reflect the results presented in Table 3. This analysis supports only an association between PNex and skin temperature, not an association with hypoxemic respiratory failure.

- If the PNex is not a validated method of measuring circadian amplitude, then the related portions of the discussion should be revised accordingly.

Figure 1

- Please provide a legend for the green, red, and black tracings. The caption says that the Y axis is hours; is this correct? (It seems like the X axis is hours?)

Figure 2

- Consider defining in the methods that you plan to compare day 1, 2 and 3 within the 5 time windows. Were any quantitative analyses applied to these 1/2/3 day comparisons? Clarifying why it was included and what these results mean would be helpful.

Reviewer #3: See attached file

**Review for PONE-D-21-34906**

In this retrospective study among 429 covid-19 patients admitted to the general ward, the authors aim to evaluate whether circadian rhythms can be observed in those patients with respect to heart rate, respiratory rate and skin temperature. Moreover, the authors explore a possible association between circadian pattern amplitude and hypoxic respiratory insufficiency. Data from five predefined time intervals were analyzed: (1) the first and (2) last three days of admission, the three days (3) preceding and (4) succeeding hypoxic respiratory insufficiency, and the (5) last three days before death.

Results revealed that heart rate and respiratory rate followed a circadian pattern in all stages of hospital admission, except for the days prior to death. Skin temperature only followed a circadian pattern on admission, discharge, and the days preceding respiratory insufficiency.

The authors conclude that these circadian patterns could improve monitoring- and alarm strategies. However, the predictive power of peak-nadir excursion however appears to be low.

The authors are to be complimented for their initiative to provide data in this important and new field of research. However, there are significant issues regarding explanation of methods, data analysis and the presentation of the results. Furthermore, part of the conclusion does not seem to be supported by the data provided.

*Major Comments*

1. We ask the authors to explain in more detail the rational for the definition of each predefined stages/cohorts? Why did the authors did not perform a longitudinal statistical analysis? It would be interesting to know if circadian patterns change over time depending on specific covariates (covariate analysis: e.g. respiratory insufficiency as a covariate). We suggest a separate statistical methods paragraph. Why are p-values not provided within the manuscript. A review by an independent statistician may be helpful.

2. The applied definition of _a circadian pattern_ should be explained in the manuscript. Sentences like _A circadian pattern was observed in HR and RR during stages I-IV._ are unclear. When should clinicians deem a pattern as a _normal circadian rhythm_? Did the authors look for indicators of circadian disturbances when analysing the data? A longitudinal analysis of patterns comparing patients with and without developing respiratory insufficiency would be very interesting for the readership.

3. The results section is very hard to "digest"; especially for readers who are not experts in the s very new field of research. We strongly suggest to amend the results section with figures, which illustrate the main results more clearly.

4. The authors conclude that circadian patterns of heart rate, respiratory rate and skin temperature could improve monitoring- and alarm strategies. At this stage of the manuscript, we do not think that this conclusion is supported by the data. We agree, that including informations of circadian profiles within in prediction models and monitoring algorithms may be an interesting and helpful approach in the future. We ask the authors to provide 1 or 2 concrete examples or clinical scenarios how this could be helpful in clinical decision making. However, the authors should be clear, that this is not supported by their data and should remove statements from the conclusion section.

5. Page 10, Line 204: „The mean circadian pattern amplitude showed differences between stages, but only an increasing PNex of skin temperature was associated with developing respiratory insufficiency and with only a small effect size." Please provide a P-value. Was the association statistical significant?

*Further Comments*

- Page 3, Line 42-44: „Since September 2020, patients with covid-19 are treated with dexamethasone, which has a suppressive effect on the circadian pattern of the human metabolism." What does _suppressive effect on circadian pattern_ mean? Please explain.

- Page, Line 226: „The decrease of circadian rhythm might be due to severe illness and extreme physical stress, but could also have been influenced by age, comorbidities and medication. Furthermore, circadian rhythms are highly influenced by light input. As part of palliative care, patients were often relocated to single rooms with closed blinds for comfort. This might have played a role in the decrease of circadian pattern seen in all vital parameters.“ Please explain in more detail why circadian rhythm disturbances can occur in the context of severe diseases, e. g. closed eyes, reduced physical activity, inflammation etc.

6. PLOS authors have the option to publish the peer review history of their article (what does this mean?). If published, this will include your full peer review and any attached files.

Reviewer #1: No

Reviewer #2: No

Reviewer #3: No

---

## [Author Response · Author response to Decision Letter 0]

23 Feb 2022

Circadian patterns of heart rate, respiratory rate and skin temperature in hospitalized covid-19 patients - Response to reviewers’ and editors’ comments

Dear prof. dr. Oster, dear reviewers,

We would like to thank you for considering our manuscript for possible publication, and for the opportunity to improve our work using the detailed and specific reviewers’ comments. We noticed that all reviewers asked for substantial changes in the analytical approach and we have adjusted our analyses and the Methods section accordingly. Several reviewers suggested to use a cosinor model instead of peak-nadir excursion, and we believe the use of this type of modelling has improved our manuscript substantially. We have also redefined our cohorts, to avoid both bias and confusion. We decided to keep patients with respiratory insufficiency and patients who died separate, since we feel that the clinical circumstances of these patients were very different. 

Because of the substantial revision of methodology and statistical analysis, some sections of the manuscript have been rewritten. We aimed to address as many concerns as possible, but you will find that in the new manuscript some comments are no longer applicable. 

We believe that these changes have considerably improved the manuscript. If there are additional comments and required revisions, we will, of course, be happy to address each of these. 

Looking forward to hearing from you, 

Sincerely,

Harriët van Goor 

h.m.r.vangoor-3@umcutrecht.nl

 

EDITOR

Fix issues with citations.

We have critically revised our references, and have adjusted citations in accordance with the suggestions made by the reviewers. 

Clarify information on dexamethasone treatment and absent data.

We have added information on dexamethasone treatment, and have provided information on the amount of continuous data used in analysis.

Provide more details on methodology and statistics as specified by the reviewers.

We have considerably modified the analysis methodology as per the reviewers’ suggestions, and have added more elaborate explanations and references to substantiate our choices. 

Improve discussion about alternative methodologies and include more recent references.

We have added references to the new methodological approach, which was recently been developed and has been used to assess circadian rhythm in HRV in covid-19 patients. We have added several more recent references in the discussion, especially regarding covid-19. 

Consider including a study population that did not develop hypoxemic respiratory insufficiency – or at least discuss this omission and tone down conclusions.

We have redefined our cohorts to clarify the distinction between patients that did and did not experience respiratory insufficiency. 

Consider including a flow diagram to describe the total number of potentially eligible patients, and those excluded at each stage (e.g. as suggested by the STROBE statement).

We have added a flow chart (figure 1) of patient inclusion. 

Avoid including patient data in more than one time-period “cohort” and then comparing differences between time 

We have redefined our cohorts to clarify the distinction between patients that did and did not experience respiratory insufficiency, and have aimed to model coefficient changes over time. 

If no cosinor analysis is applied, thenit seems like this peak-nadir measurement (PNex) will only provide the range of daytime vs. nighttime data. Please include references for using this measurement in circadian analyses. 

We have no longer used the method of PNex, but used cosinor modelling as suggested by several reviewers. 

Include a quantitative analysis for rhythmicity.

We have added a rhythmicity test based on the goodness of fit of the cosinor model for the population mean cosinor models. 

Discuss more specifically how findings could be translated into the clinics.

We have specifically addressed the implications for both predictive modelling and clinical practice. 

 

Reviewer #1 

1. Introduction: “Since September 2020, patients with covid-19 are treated with dexamethasone[15], which has an suppressive effect on the circadian pattern of the human metabolism[16].” The citation (16) is not applicable. In the cited study dexamethasone was administered in the afternoon. Within the clinical routine, dexamethasone is likely to be administered in the early morning which might rather result in a strengthening of circadian rhythms. How many patients received dexamethasone in this study? Why September 2020, not June 2020?

Thank you very much for this observation. We did not realize the difference in timing of administration of dexamethasone would have the opposite impact in our study population. We have adjusted the sentence to better reflect this. As for the amount of patients receiving dexamethasone, we have added this to our baseline table. We chose September 2020 since this was the date of the first scientific publication of the RECOVERY trial according to the website of the RECOVERY trial. We have adjusted this to July 2020 when the first preliminary report was published by the NEJM. 

2. Methods: “Inclusion of patients stopped because the wearable sensor was no longer available.” Please specify: Was the availability of the sensor tied to funding for the study or did the company stop production due to unreliable measurement accuracy?

Unfortunately, the reason behind the availability of the sensor is too long a story to add to this paper. The wearable we used is reliable (Selvaraj et al., 2018). The distributor of the wearable (Philips) however had developed their own patch sensor and stopped the production of the old sensor (the one we used). Unfortunately, that new sensor turned out to be not reliable. The distributor then decided to stop selling these sensors all together and instead focused on a new type of sensor, leaving us without sensors. We have added a summary of the reason to the paper. Hopefully this provides enough clarification. 

3. Methods: “Patients with a pacemaker did not receive a sensor since RR measurements might result unreliable in paced rhythms.” Please clarify: Were those patients completely excluded from the study or only RR measurements were excluded for those patients? 

Since patients with a pacemaker did not receive a sensor, they had no available continuous data and therefore did not meet the inclusion criteria as described in the paragraph on ‘data collection’. We have added ‘sensor’ to the notion of ‘available continuous data’ to clarify that only patients who received a sensor could be included. 

4. Methods: „Since our cohort included dying patients, we used wide limits for improbably 114 data (for RR <1/min & >80/min; for HR <30/min & >280/min; for sT < 25°C).” The lower limits for temperature and respiratory rate are extremely wide and should be critically revised. 

Thank you for this feedback, we have discussed this extensively. The limits were chosen based on literature and clinical experience, but also on observations in the data. Especially in patients who were monitored while dying, it is hard to say which values are still valid and which are not. Therefore we have chosen to use wide limits, but also use additional methods to filter out possible artefacts (by deleting short lasting changes in HR and RR, and the ‘warming up’ period of skin temperature), and using a median filter of 15 minutes, before performing further analyses. 

5. Methods: “Data was divided in five cohorts based on different stages of disease…. Hypoxic respiratory insufficiency was defined as the need for 15 l/min oxygen therapy.” Why did the authors not stratify according to the WHO criteria in mild, intermediate and severe COVID19. Please discuss and reference, if this method has been used before. 

We did no use the WHO criteria for multiple reasons: first of all, patients with mild disease are not admitted to the hospital in the Netherlands, leaving us with only two categories. Secondly, the difference between moderate and severe disease according to the WHO guidelines is made based on respiratory rate (with a threshold of 30/min) and oxygen saturation (with a threshold of 90% on room air). Due to the nature of our data, which includes continuous respiratory rate values, we have observed that respiratory rate changes often and quickly. A patient could change between the moderate and severe group multiple times within one hour. Regarding the threshold for oxygen saturation, this threshold was made to use without room air. The vast majority of patients in our cohort however receive supplemental oxygen, so it would be almost impossible to decide in which category they belong. We could have used the situation at admission, however this does not reflect the aim of our study: to determine the change in circadian pattern during deterioration. The more conventional method to define deterioration would be intensive care admission. However, we do not believe that the time of admission to the intensive care adequately reflects the moment a patient becomes respiratory insufficient. A patient could have received 100% supplemental oxygen for hours without improvement of the situation, which unfortunately happened often during the covid-19 crisis. Furthermore, many patients had treatment restrictions, and 15L/min O2 was the highest level of care they were willing to receive or which was medically responsible. 

Since we had such a specific aim and population, we decided to define a new endpoint. We hope this explanation clarifies our choice. 

6. Methods: “For quantitative assessment we divided the data in daytime (06:00-00:00) and nighttime (00:00-06:00).” The daytime period is proportionally much longer than the nighttime. Please discuss and reference, why this method was chosen and if this method has been published before. 

Thank you for this observation. Due to the new methodology, we have no longer defined daytime and nighttime. 

7. Methods: Data collection included the Charlson Comorbidity Index, but the group differences were not further discussed within the manuscript. The use of further disease severity scores for ICU patients (SOFA, GCS) would complement the author’s analysis. 

Thank you for this suggestion. Since our population is not an ICU population, but patients at the general ward, we have no routinely collected data on SOFA or GCS. Unfortunately, we also have not collected any other form of disease severity score. Instead we have tried to stratify our analysis based on disease severity: a cohort of patients who recovered without respiratory insufficiency, a cohort of patients who developed respiratory insufficiency, and a cohort of patients who died. 

8. Statistics: Although the authors nicely removed several abrupt deviations before analysis, the use of PNex measurement might not be picking up circadian trends very well. It is correlated, however results are very noisy. For a better understanding the analysis needs to be discussed and compared with other methods applied in previous publications examining vital signs. Why did the authors not perform a regular rhythmicity analysis and sine curve fit to better estimate the amplitude?

Thank you very much for this helpful suggestion. Based on this comment and comments by the other reviewers, we have chosen to change our methodology altogether. We have fitted both a population mean cosinor model and a mixed-effect cosinor model to answer our different research questions. We hope our new methodology leads to better understanding. 

9. Discussion: Studying a clinical cohort of Covid-19 patients exclusively, the authors conclude that a general knowledge of circadian patterns might improve general monitoring- and alarm strategies. This affirmation is not supported by data including other disease entities and therefore needs to be discussed including more recent literature and trials examining circadian patterns in clinical cohorts. 

Daou M, Telias I, Younes M, Brochard L, Wilcox ME. Abnormal Sleep, Circadian Rhythm Disruption, and Delirium in the ICU: Are They Related? Front Neurol. 2020 Sep 18;11:549908. doi: 10.3389/fneur.2020.549908. PMID: 33071941

Lachmann G, Ananthasubramaniam B, Wünsch VA, Scherfig LM, von Haefen C, Knaak C, Edel A, Ehlen L, Koller B, Goldmann A, Herzel H, Kramer A, Spies C. Circadian rhythms in septic shock patients. Annals of Intensive Care. 2021 11: 64. PMID 33900485

Maas MB, Lizza BD, Abbott SM, Liotta EM, Gendy M, Eed J, Naidech AM, Reid KJ, Zee PC. Factors Disrupting Melatonin Secretion Rhythms During Critical Illness. Crit Care Med. 2020 Jun;48(6):854-861. PMID: 32317599

Maas MB, Iwanaszko M, Lizza BD, Reid KJ, Braun RI, Zee PC. Circadian Gene Expression Rhythms During Critical Illness. Crit Care Med. 2020 Dec;48(12):e1294-e1299. PMID: 33031153

Thank you very much for the suggested literature. We have used several to clarify and improve the discussion of our article. However, the suggested articles are all studies on critically ill patients in the intensive care unit. We feel strongly that the vital signs of critically ill patients at an intensive care unit cannot be compared to the vital signs of moderately ill patients at a low care ward, and that monitoring strategies for both wards are different, so we have avoided direct comparison. We did add several references of studies on general wards where wearables are used for continuous monitoring of covid-19 patients. Furthermore, we have highlighted why awareness of circadian patterns might aid interpretation of vital sign trends in clinical practice. The cited paper by van Rossum et. al. ((van Rossum et al., 2021) reference [24]), who found that the alarm strategy at a surgical ward could be improved by using different threshold for days and nights, underlines the potential of knowledge of circadian patterns to improve alarm strategies.  

Reviewer #2

The main issue is that the current study design does not support an analysis of risk for hypoxemic respiratory insufficiency as the authors propose. Evaluating different time periods a single cohort of patients, comparing a pre-hypoxemic “control” period and post-hypoxemic “outcome” period in those who developed hypoxemic respiratory insufficiency (defined in this study as >15L O2 therapy), we are able to establish only that these measurements differ in these two periods of time within these patients. To evaluate for risk of hypoxemic respiratory insufficiency, the authors must include a study population that did not develop hypoxemic respiratory insufficiency. This could be done in a variety of ways to create either a cohort or a case-control study design. The analysis should be revised to include such a cohort, or the manuscript needs to be rewritten with this significant change in mind. It would also be helpful to include references using the methods employed (daytime peak vs. nighttime nadir of physiologic data) to evaluate circadian patterns in the data. 

Thank you for pointing out this issue of the study design. Based on this comment, and comments made by the other reviewers, we have decided to apply a new analysis. We have included 3 cohorts: patients who recovered without experiencing respiratory insufficiency, patients who developed respiratory insufficiency and patients who died. We have not gone as far as to make a predictive model. We feel like this study is an explorative precursor study to see is a predictive model based on circadian pattern amplitude would be feasible. To better reflect this, we have adjusted the statement of objectives in the introduction. 

Since we have abandoned the method of daytime peak and nighttime nadir, we have not included these references in the article. 

Abstract

- Please define “delta day” as a measurement in the abstract. 

- Please revise to more accurately describe the logistic regression results (that VS changes are associated with worsened hypoxemia within in this select cohort of patients; they do not demonstrate risk of developing hypoxemia – as above, this cannot be evaluated unless the study also includes patients who did not develop hypoxemic respiratory insufficiency.)

Since we have abandoned this method, the mentioned sentences are no longer part of the abstract. 

Methods

- p5. Please clarify - if sT measurements were not directly available then how were they studied? 

Skin temperature values were not available for clinical use. They were however stored for research purposes.

- Please include how many COVID19 patients were excluded for lack of sensor data.

We have added a flowchart of the patient inclusion (figure 1). We hope this will clarify the number of patients excluded during each stage.

- p.6. Please correct “improbably data.” 

We have corrected improbably to improbable.

- The authors excluded patients with < 48 hours of continuous data during a three-day period, except for the post-hypoxemic respiratory failure period. This suggests that patients who developed hypoxemic respiratory failure very quickly (eg after 1-2 days) or who died shortly after admission without developing hypoxemic respiratory failure, etc. would have been excluded. This biases the study population to those patients who had a relatively slow progression of disease and may bias the study population away from those admitted to the ICU (it is unclear whether remote monitoring continued after ICU admission?). The authors should consider revising the analysis, using a shorter time window for each “period”. 

Thank you for this helpful comment. We have redefined our cohorts, and have limited the amount of needed data to 4-hours. We believe this reduces the introduced bias. To be able to look at changes over time, we did include a three-day period of data. 

- Consider including a flow diagram to describe the total number of potentially eligible patients, and those excluded at each stage (e.g. as suggested by the STROBE statement). 

Thank you very much for this suggestion, we have included a flow chart (figure 1). 

- Including patient data in more than one time-period “cohort” and then comparing differences between time periods is problematic from a study design perspective. This introduces bias towards the null – which makes it more difficult to identify true differences in vital signs between cohorts. The authors should consider revising their compared time periods so that no overlap occurs. 

Thank you for this observation. As mentioned before, we have redefined the cohorts, limiting the introduced bias. 

- How accurately was the onset of hypoxemic respiratory insufficiency defined? It would be helpful to describe how this information was obtained, e.g. if this was time stamped (hour and minute) to ensure that the vital sign data was assigned correctly as pre- vs. post-intervention. 

The onset of respiratory insufficiency was defined as the first recording in the electronic patient record of either 15L/min oxygen therapy, high flow oxygen therapy, ventilation, or a cardiac arrest call. We have added this information to the data selection paragraph of the methods. 

- p.7. Were peaks and nadirs drawn from raw data? Typically, circadian analyses using peak/nadir are drawn from a cosinor or sinusoidal analysis that accounts for the entire dataset shifting + and – in a circadian fashion, which is less sensitive to random noise (e.g., Shoben Am J Epi 2011 for an excellent example.) If no cosinor analysis is applied, thenit seems like this peak-nadir measurement (PNex) will only provide the range of daytime vs. nighttime data. Please include references for using this measurement in circadian analyses. 

Thank you very much fort his comment. It has led us to revise our methodology, and we have now fitted a cosinor analysis (including references). 

Results:

Thank you very much for the following comments regarding the results section. Since we have changed the methodology and therefore the results, these comments are no longer applicable. We have however tried to apply the feedback in general to the new result section. 

- p. 7. Please rename the 5 time periods; they are not truly “cohorts” as they include the same patients, and the patient data is actually duplicated across some of these analytic groups. Suggest using “time periods” or something similar. 

- How do more patients have respiratory insufficiency than deterioration? This might be clarified by using more distinct/separated definitions of time periods for comparison and/or a flow diagram as suggested above. 

- Table 3 sugests that day 1 vs. 2 of respiratory insufficiency and day 1 vs 3 of respiratory insufficiency were compared in logisitc regression. This is different from the analyses described in the Methods. Please clarify. 

-p.9; was mean PNex compared quantitatively between stages using a statistical test? If so, consider adding these comparisons to table 2. 

-p.9, lines 189-191 and 191-192 should be edited for accuracy. Since patients without hypoxemic respiratory insufficiency were not included in this analysis, we cannot evaluate whether this development was associated with the development of hypoxemia. All we know is that the delta PNex was associated with skin temperature in this select cohort of patients who will go on to develop respiratory insufficiency. 

-Table 3: Table 3 compares the difference in Pnex between days 1 and 2 and days 1 and 3. Please clarify – is this limited to stage III patients? If so, please add this detail and the number of patients included to the title. 

Discussion:

- I agree that Figure 1 suggests qualitatively that there is circadian variability given the cyclical nature of overall VS pattern. However, it’s hard to claim definitively that there is no circadian variability based on a subjective assessment (eg for the patients with mortality.) From figure 1, stage I and stage III do not necessarily look all that different – it is not clear what the authors used as a cut off for circadian rhythm presence vs. absence. If a quantitative analysis is possible, one should be included. If not, the discussion should reflect this uncertainty throughout. 

Thank you for this suggestion. We have added a quantitative analysis of rhythmicity to the methods of our study. 

- The Discussion should be edited in several places to accurately reflect the results presented in Table 3. This analysis supports only an association between PNex and skin temperature, not an association with hypoxemic respiratory failure. 

Since we have changed this analysis, these results are no longer part of the study.

- If the PNex is not a validated method of measuring circadian amplitude, then the related portions of the discussion should be revised accordingly.

Although other studies did use PNex to model circadian rhythms ((Davidson et al., 2021) reference [21]), we have decided not to use PNex as method but used a cosinor model instead. We have added to the discussion that the different methods might have different results. 

Figure 1

- Please provide a legend for the green, red, and black tracings. The caption says that the Y axis is hours; is this correct? (It seems like the X axis is hours?)

Indeed, the X-axis was supposed to be hours, thank you for pointing this out. We have added the legend to the description of the figure. 

Figure 2

- Consider defining in the methods that you plan to compare day 1, 2 and 3 within the 5 time windows. Were any quantitative analyses applied to these 1/2/3 day comparisons? Clarifying why it was included and what these results mean would be helpful.

Since we have changed this analysis, these comparisons are no longer part of the study.

 

Reviewer #3 

*Major Comments*

1. We ask the authors to explain in more detail the rational for the definition of each predefined stages/cohorts? Why did the authors did not perform a longitudinal statistical analysis? It would be interesting to know if circadian patterns change over time depending on specific covariates (covariate analysis: e.g. respiratory insufficiency as a covariate). We suggest a separate statistical methods paragraph. Why are p-values not provided within the manuscript. A review by an independent statistician may be helpful. 

Thank you for this valuable comment. We redefined the cohorts, and have paid attention to explaining why the new cohorts were chosen. Using the mixed-effect cosinor model, we have aimed to gain insight into changes of amplitude over time depending on the cohort (figure 3). Although this analysis does not provide prognostic information, we believe it does explore differences in circadian patterns over the days that might be used as input for prognostic studies. 

2. The applied definition of _a circadian pattern_ should be explained in the manuscript. Sentences like _A circadian pattern was observed in HR and RR during stages I-IV._ are unclear. When should clinicians deem a pattern as a _normal circadian rhythm_? Did the authors look for indicators of circadian disturbances when analysing the data? A longitudinal analysis of patterns comparing patients with and without developing respiratory insufficiency would be very interesting for the readership.

Thank you for this feedback. We have chosen to change the method altogether, to avoid confusing and unclarity. We have added a quantitative measure of rhythmicity by doing a ‘goodness of fit’ test of the cosinor model. We have aimed to show longitudinal changes by using the mixed-effect cosinor model. 

3. The results section is very hard to "digest"; especially for readers who are not experts in the s very new field of research. We strongly suggest to amend the results section with figures, which illustrate the main results more clearly.

Thank you very much for this suggestion. Since the methods have changed, the results section has changed too. We have added figure 3 and supplemental figure 1 to illustrate results. 

4. The authors conclude that circadian patterns of heart rate, respiratory rate and skin temperature could improve monitoring- and alarm strategies. At this stage of the manuscript, we do not think that this conclusion is supported by the data. We agree, that including informations of circadian profiles within in prediction models and monitoring algorithms may be an interesting and helpful approach in the future. We ask the authors to provide 1 or 2 concrete examples or clinical scenarios how this could be helpful in clinical decision making. However, the authors should be clear, that this is not supported by their data and should remove statements from the conclusion section.

We agree that the improvement of monitoring- and alarm strategies was not the scope of this article and have removed this statement from the conclusion. In the paragraph ‘use in predictive modelling and clinical practice’ you will find a concrete example of how awareness of circadian patterns might influence clinical decision making. 

5. Page 10, Line 204: „The mean circadian pattern amplitude showed differences between stages, but only an increasing PNex of skin temperature was associated with developing respiratory insufficiency and with only a small effect size." Please provide a P-value. Was the association statistical significant? 

We have changed the analysis and provided p-values in the new result section. 

*Further Comments*

- Page 3, Line 42-44: „Since September 2020, patients with covid-19 are treated with dexamethasone, which has a suppressive effect on the circadian pattern of the human metabolism." What does _suppressive effect on circadian pattern_ mean? Please explain.

We agree with you and reviewer 1 that this statement is unclear and incorrect. We have changed the sentence to ‘..dexamethasone, which can affect the circadian pattern of the human metabolism depending on time of administration’ to better reflect that dexamethasone can either suppress or amplify circadian patterns, depending on the parameter and the timing of administration.

- Page, Line 226: „The decrease of circadian rhythm might be due to severe illness and extreme physical stress, but could also have been influenced by age, comorbidities and medication. Furthermore, circadian rhythms are highly influenced by light input. As part of palliative care, patients were often relocated to single rooms with closed blinds for comfort. This might have played a role in the decrease of circadian pattern seen in all vital parameters.“ Please explain in more detail why circadian rhythm disturbances can occur in the context of severe diseases, e. g. closed eyes, reduced physical activity, inflammation etc.

Thank you for this suggestion. We have clarified these statements with details and added several references. 

 

References

Davidson, S., Villarroel, M., Harford, M., Finnegan, E., Jorge, J., Young, D., Watkinson, P., & Tarassenko, L. (2021). Day-to-day progression of vital-sign circadian rhythms in the intensive care unit. Critical Care. https://doi.org/10.1186/s13054-021-03574-w

Selvaraj, N., Nallathambi, G., Moghadam, R., & Aga, A. (2018). Fully Disposable Wireless Patch Sensor for Continuous Remote Patient Monitoring. Proceedings of the Annual International Conference of the IEEE Engineering in Medicine and Biology Society, EMBS. https://doi.org/10.1109/EMBC.2018.8512569

van Rossum, M. C., Vlaskamp, L. B., Posthuma, L. M., Visscher, M. J., Breteler, M. J. M., Hermens, H. J., Kalkman, C. J., & Preckel, B. (2021). Adaptive threshold-based alarm strategies for continuous vital signs monitoring. Journal of Clinical Monitoring and Computing. https://doi.org/10.1007/s10877-021-00666-4

---

## [Decision Letter · Decision Letter 1]

11 Apr 2022

PONE-D-21-34906R1Circadian patterns of heart rate, respiratory rate and skin temperature in hospitalized covid-19 patientsPLOS ONE

Dear Dr. van Goor,

Thank you for submitting your manuscript to PLOS ONE. After careful consideration, we feel that it has merit but does not fully meet PLOS ONE’s publication criteria as it currently stands. Therefore, we invite you to submit a revised version of the manuscript that addresses the points raised during the review process.

We look forward to receiving your revised manuscript.

Kind regards,

Henrik Oster, Ph.D.

Academic Editor

PLOS ONE

Journal Requirements:

Reviewers' comments:

Reviewer's Responses to Questions

**Comments to the Author**

1. If the authors have adequately addressed your comments raised in a previous round of review and you feel that this manuscript is now acceptable for publication, you may indicate that here to bypass the “Comments to the Author” section, enter your conflict of interest statement in the “Confidential to Editor” section, and submit your "Accept" recommendation.

Reviewer #1: All comments have been addressed

Reviewer #2: (No Response)

2. Is the manuscript technically sound, and do the data support the conclusions?

Reviewer #1: Yes

Reviewer #2: Yes

3. Has the statistical analysis been performed appropriately and rigorously? 

Reviewer #1: Yes

Reviewer #2: Yes

4. Have the authors made all data underlying the findings in their manuscript fully available?

Reviewer #1: Yes

Reviewer #2: No

5. Is the manuscript presented in an intelligible fashion and written in standard English?

Reviewer #1: Yes

Reviewer #2: Yes

6. Review Comments to the Author

Reviewer #1: This reviewer is grateful that the authors addressed all issues raised.

This reviewer is grateful that the authors addressed all issues raised.

Now, the minimum character count is met......

Reviewer #2: PONE-D_21_34906_R1

Review 4/4/2022

The authors have substantially revised their manuscript reflecting a complete revision of their analysis, now comparing circadian patterns in vital signs between three outcome-based groups: patients who recovered without respiratory insufficiency, those who developed respiratory insufficiency, and those who died. This analysis is much clearer and more readily interpreted. I have one remaining question regarding time frame definitions for analysis within each cohort, as well as some minor suggestions/clarifications.

Introduction

1. P.4 – This paragraph could likely be shortened without losing substantive information. For example, “Assuming a physiological difference in vital sign values between night and day might be closer to reality than assuming equal values throughout the entire 24-hour cycle” may no longer fit the analysis employed in this manuscript.

2. p.4 – minor detail, consider adding the 3rd cohort (patients who did not develop respiratory insufficiency) to the description of your 2nd research aim.

Methods

1. p.6 – Please clarify the sentence beginning “Patients were included if…”. The “respiratory insufficiency resp. death” is confusing.

2. Suggest moving the “4 hours of data” requirement to earlier in the paragraph – you restricted this analysis to patients who had at least 4 hours of (continuous?) data within the time frame of interest.

3. Because you analyzed serial days within the 72h window per patient, did you require 4 hours of data per day?

4. How were you able to do the cosinor modeling for patients with only 4h of data per day? It seems like this would require more data.

5. Choosing comparable time frames in these three outcome-defined cohorts is challenging. The authors have chosen three different time frames during the admission, in which the patient physiology could be expected to be very different – 72h after admission for patients who recovered, 72h before respiratory failure in those patients, and 72h before death in those patients. Because LOS in the different cohorts was quite different, to help understand the “comparability” of these patients, providing some information about where this window fell in the average length of stay for each cohort would be helpful. (For example – the 72h is the first 3 days out of an average 6 day LOS for the recovery cohort. But was it also on average the first 3 days for the respiratory failure cohort – e.g., when in the hospital stay did the respiratory failure occur?) This matters because patients who have been hospitalized longer have more opportunity for circadian disruption, and you may be finding changes in this cohort that are attributable solely to longer LOS/later time of evaluation and not attributable to clinical decline. If this is the case, this source of bias would need to be explicitly discussed in the discussion (and could be addressed in multiple locations in the discussion, as it may help explain the inconsistent findings between the respiratory failure and mortality cohorts, as well as the presence of respiratory and temperature variability only in the recovery cohort.) One could also consider trying to address this analytically, e.g. by matching recovery patients to patients in the other cohorts based on LOS, but this is probably not worth doing.

Results

1. p.10 - Please clarify “stratified by day”. Do you mean stratified by cohort (as in the title of figure 3)?

Minor comments

- Some editing for grammar would be helpful, e.g. p. 3 “circadian rhythm become increasingly more pronounced”, p.4 “three…research questions”, etc.

7. PLOS authors have the option to publish the peer review history of their article (what does this mean?). If published, this will include your full peer review and any attached files.

Reviewer #1: No

Reviewer #2: No

---

## [Author Response · Author response to Decision Letter 1]

19 Apr 2022

Response to reviewers

The authors have substantially revised their manuscript reflecting a complete revision of their analysis, now comparing circadian patterns in vital signs between three outcome-based groups: patients who recovered without respiratory insufficiency, those who developed respiratory insufficiency, and those who died. This analysis is much clearer and more readily interpreted. I have one remaining question regarding time frame definitions for analysis within each cohort, as well as some minor suggestions/clarifications.

Introduction

1. P.4 – This paragraph could likely be shortened without losing substantive information. For example, “Assuming a physiological difference in vital sign values between night and day might be closer to reality than assuming equal values throughout the entire 24-hour cycle” may no longer fit the analysis employed in this manuscript.

Thank you for the suggestion, we have critically reviewed the paragraph and have removed several sentences.

2. p.4 – minor detail, consider adding the 3rd cohort (patients who did not develop respiratory insufficiency) to the description of your 2nd research aim.

We have added the third cohort to the description of the research aim.

Methods

1. p.6 – Please clarify the sentence beginning “Patients were included if…”. The “respiratory insufficiency resp. death” is confusing.

We have clarified and elaborated on the choice of data for patients in each cohort. 

2. Suggest moving the “4 hours of data” requirement to earlier in the paragraph – you restricted this analysis to patients who had at least 4 hours of (continuous?) data within the time frame of interest.

We decided to write down the methods in the same order as we analysed the data, to make reproduction of the study as easy as possible. Since we first selected the 72 hour timeframes, and later excluded those patients who appeared to have less than 4 hours of data within this timeframe, this is the chronological order of analysis. 

3. Because you analyzed serial days within the 72h window per patient, did you require 4 hours of data per day?

No, we required 4 hours of data in total to include a patient in the population mean cosinor and rhythmicity test. For the stratification per day, a patient was only included in the day he/she had data during that day. 

4. How were you able to do the cosinor modeling for patients with only 4h of data per day? It seems like this would require more data.

The cosinor model does not need a full cycle to estimate a cosinor curve. 4 hours of data, plus the indication where on the 24 hour cycle this data is located (the time) was enough to estimate the rest of the curve. 

5. Choosing comparable time frames in these three outcome-defined cohorts is challenging. The authors have chosen three different time frames during the admission, in which the patient physiology could be expected to be very different – 72h after admission for patients who recovered, 72h before respiratory failure in those patients, and 72h before death in those patients. Because LOS in the different cohorts was quite different, to help understand the “comparability” of these patients, providing some information about where this window fell in the average length of stay for each cohort would be helpful. (For example – the 72h is the first 3 days out of an average 6 day LOS for the recovery cohort. But was it also on average the first 3 days for the respiratory failure cohort – e.g., when in the hospital stay did the respiratory failure occur?) This matters because patients who have been hospitalized longer have more opportunity for circadian disruption, and you may be finding changes in this cohort that are attributable solely to longer LOS/later time of evaluation and not attributable to clinical decline. If this is the case, this source of bias would need to be explicitly discussed in the discussion (and could be addressed in multiple locations in the discussion, as it may help explain the inconsistent findings between the respiratory failure and mortality cohorts, as well as the presence of respiratory and temperature variability only in the recovery cohort.) One could also consider trying to address this analytically, e.g. by matching recovery patients to patients in the other cohorts based on LOS, but this is probably not worth doing.

Thank you for your thorough consideration of the implications of timeframe selection. We share your opinion that the comparability of patients would be less if the timeframes were selected from different parts of the admission. Because respiratory insufficiency usually occurs within the first 72 hours of admission (median 33 hours), we chose the first 3 days of admission for the recovery (“control”) group too. We realise we failed to mention this in earlier versions of the manuscript, but we have now added it to the methods on page 6/7 of the manuscript. 

Regarding the mortality cohort, mortality usually occurred later than 72 hours after admission. The comparability of this cohort with the other two cohorts could indeed be questioned. We have added this to the discussion on page 13 and to the limitations of the study on page 14, including the consideration to use a case-control design matched on length of hospitalization in future research. 

Results

1. p.10 - Please clarify “stratified by day”. Do you mean stratified by cohort (as in the title of figure 3)?

Actually, figure 3 presents the cosinor characteristics stratified by cohort and by day. We have clarified this both in the manuscript and in the title of figure 3. 

Minor comments

- Some editing for grammar would be helpful, e.g. p. 3 “circadian rhythm become increasingly more pronounced”, p.4 “three…research questions”, etc.

Thank you for your acuity. We have checked the entire manuscript and have corrected several grammatical mistakes.

---

## [Editor Report · Decision Letter 2]

22 Apr 2022

Circadian patterns of heart rate, respiratory rate and skin temperature in hospitalized COVID-19 patients

PONE-D-21-34906R2

Dear Dr. van Goor,

We’re pleased to inform you that your manuscript has been judged scientifically suitable for publication and will be formally accepted for publication once it meets all outstanding technical requirements.

Kind regards,

Henrik Oster, Ph.D.

Academic Editor

PLOS ONE

Additional Editor Comments (optional):

n/a
---

## [Editor Report · Acceptance letter]

28 Jun 2022

PONE-D-21-34906R2 

Circadian patterns of heart rate, respiratory rate and skin temperature in hospitalized COVID-19 patients 

Dear Dr. van Goor:

I'm pleased to inform you that your manuscript has been deemed suitable for publication in PLOS ONE. Congratulations! Your manuscript is now with our production department. 

Kind regards, 

on behalf of

Prof. Henrik Oster 

Academic Editor

PLOS ONE